# Diffusion-Slip Boundary Conditions for Isothermal Flows in Micro- and Nano-Channels

**DOI:** 10.3390/mi13091425

**Published:** 2022-08-29

**Authors:** Alwin Michael Tomy, S. Kokou Dadzie

**Affiliations:** School of Engineering and Physical Sciences, Heriot-Watt University, Edinburgh EH11 4AS, UK

**Keywords:** diffusion slip, volume diffusion, micro- and nano-channels

## Abstract

Continuum description of flows in micro- and nano-systems requires ad hoc addition of effects such as slip at walls, surface diffusion, Knudsen diffusion and others. While all these effects are derived from various phenomenological formulations, a sound theoretical ground unifying these effects and observations is still lacking. In this paper, adopting the definition and existence of various type of flow velocities beyond that of the standard mass velocity, we suggest derivation of model boundary conditions that may systematically justify various diffusion process occurring in micro- and nano-flows where the classical continuum model breaks down. Using these boundary conditions in conjunction with the classical continuum flow equations we present a unified derivation of various expressions of mass flow rates and flow profiles in micro- and nano-channels that fit experimental data and provide new insights into these flow profiles. The methodology is consistent with recasting the Navier–Stokes equations and appears justified for both gas and liquid flows. We conclude that these diffusion type of boundary conditions may be more appropriate to use in simulating flows in micro- and nano-systems and may also be adapted as boundary condition models in other interfacial flow modelling.

## 1. Introduction

Advancements in fabrication methods and developments in microfluidics are driving the research interests to understand the unconventional physics that govern the operation and manufacturing of micro- and nano- scale devices such as micropumps, heat exchangers for electronic devices, gas chromatography analyzers and other micro-electro-mechanical-systems [1,2,3,4,5]. Classical flow models such as the Navier–Stokes (N-S) equations with no-slip boundary condition are unable to replicate some of the phenomena pertaining fluid flows in the micro- and nano- length scales [6]. Experimentally, it has been observed and confirmed that there is a significant enhancement of fluid flow through micro- and nano- channels but classical continuum models fail to accurately predict the observed flow enhancement as well as other flow profiles such as non-linear pressure profiles [7,8,9]. Initial observations were made by Knudsen while investigating rarefied gas flows through narrow tubes [10]. Since then several studies have verified this enhancement for gas flows through micro-channels at various states of rarefaction [11,12,13] and for liquid flows through nano-tubes [7,8,14,15]. The degree of rarefaction in a gas is defined by its Knudsen number, Kn, as the ratio between the mean free path (λ) of the gas particles to a characteristic length of the flowing system [16]. Based on the Knudsen number, gas flows are generally divided into four regimes: the continuum flow regime with Kn≤0.001, the slip flow regime with 0.001≤Kn≤0.1, the transition regime with 0.1≤Kn≤10, and the free molecular regime with Kn≥10. In an effort to extend the application of continuum flow models to predict non-conventional fluid flow phenomena in the rarefied regime, many researchers have over the years formulated slip boundary conditions and new continuum models e.g. Extended Navier–Stokes equations (ENSE) [17,18], Bi-velocity model [19] and Recast Navier–Stokes (RNS) equations [20,21]. Of the velocity boundary conditions developed, the most elegant and widely used is the Maxwell slip model, introduced by James Clerk Maxwell in 1879 [22]. Arkilic et al. [11] analytically solved a simplified set of Navier–Stokes equations along with first-order Maxwell slip boundary conditions for micro-channel flow and successfully predicted the mass flow rate up to Kn ∼0.1. For high values of Knudsen number, up to Kn ∼1.2, the validity of N-S equations to predict these flow rates is seen to be improved by using a second-order slip boundary conditions with an appropriate selection of slip coefficients extracted from the given experimental data [23,24]. Karniadakis and Beskok [1] proposed a general slip boundary conditions, and their solution of N-S equations for a micro-channel case agrees well up to Kn =5.

The concept of a diffusive velocity or a diffusive flux based on Fick’s law has given rise to diffusive slip boundary condition and new continuum models such as ENSE, Bi-velocity model and Recast Navier–Stokes model. Adachi et al. [25] introduced a diffusion based velocity-slip boundary condition that is based on physical insights to predict the diffusive velocity at the wall. Veltzke and Thöming [26] have shown that a superposition of convective transport and Fickian diffusion term matches experimental data for micro-channel flows, up to Knudsen number of 0.4. Dongari et al. [27] provided analytical solution for the ENSE model applied for a micro-channel flow and validated the solutions against Direct Simulation Monte Carlo (DSMC) solutions, showing a good agreement in the early and late transition regimes (Kn < 0.3 and 1 < Kn < 10). Brenner introduced the concept of volume diffusion hydrodynamics [6,28,29]. Dadzie and Brenner [30], on the basis of the volume diffusion hydrodynamics, derived analytical solution to micro-channel flows that agreed well with Ewart et al.’s [13] data up to Knudsen number of 5.

Motivated by the previous works, in this paper we provide a formulation of new velocity boundary conditions unifying the existing Maxwellian and diffusion velocities from recast Navier–Stokes models [20]. This new boundary conditions when used with the conventional Navier–Stokes equations, predict gas mass flow rates from the continuum to free-molecular regime. Additional terms appearing in the expressions for the flow rates directly link to the definition of the new diffusion velocities. The new boundary conditions also reveal new insights into the velocity and pressure profiles that are more consistent with the flow enhancement observed. For example, in the high Knudsen number regime, the flow profiles become clearly that of plug flows corroborating the high flow enhancement phenomena beyond the simple traditional wall slip velocity explanation.

The rest of the paper is organised as follows: In Section 2, we introduce the new diffusion-slip boundary conditions and the reduced governing equations for flows in micro-channels. Analytical solutions for micro- and nano-channel gas flows are derived in Section 2.4 and include effects of rarefaction as described in Section 2.3. In Section 3, we compare the dimensionless flow-rate predicted by the new model with experimental data and also present the solutions for velocity and pressure profiles at various states of rarefaction. Finally, in Section 4, we outline our concluding remarks based on the results of the study.

## 2. Governing Equations and the New Models

The present governing equations and models are based on the observation that a change in velocity variable in classical Navier–Stokes equations leads to various type of mass and volume diffusion continuum models [20].

### 2.1. Governing Equations

We consider the compressible Navier–Stokes set of equations to model flow through the micro-channels. In the absence of any temperature gradients the Navier–Stokes system of equations, ignoring the body force, for single species can be written in Cartesian coordinates as:(1)$$\begin{array}{ccc} \frac{\partial \rho }{\partial t}+\frac{\partial (\rho U_{i})}{\partial x_{i}} & = & \;0, \end{array}$$(2)$$\begin{array}{ccc} \rho \left(\frac{\partial U_{j}}{\partial t}+U_{i}\frac{\partial U_{j}}{\partial x_{i}}\right) & = & -\frac{\partial P}{\partial x_{j}}-\frac{\partial \tau_{ij}}{\partial x_{i}}, \end{array}$$
where *i* and *j* are the coordinate indices and ρ, *U*, *P* and τ are the density, velocity, pressure and viscous stress, respectively. For a newtonian fluid,
(3)$$\begin{array}{c} \tau_{ij}=-\mu \left(\frac{\partial U_{j}}{\partial x_{i}}+\frac{\partial U_{i}}{\partial x_{j}}\right)+\frac{2}{3}\mu \delta_{ij}\frac{\partial U_{k}}{\partial x_{k}}, \end{array}$$
where μ is the dynamic viscosity and $\delta_{ij}$ is the Kronecker Delta function, and equation of state for an ideal gas is given by,
(4)$$\begin{array}{c} P=\rho RT, \end{array}$$
where R is the specific gas constant and *T* is the temperature. In this paper we consider an isothermal pressure driven flow of an ideal gas through a rectangular micro-channel of length *L*, width *w* and height *h*, as shown in Figure 1. The stream-wise coordinate is *x* with velocity $U_{x}=u$, the wall-normal coordinate is *y* with velocity $U_{y}=v$, and the height to length ratio is ϵ=h/L. For the case of ϵ≪1 and h/w≪1 we can consider the flow to be two-dimensional, neglecting the variations in the z direction.

Arkilic et al. [11] in his work on slip flows in micro-channels has categorised the flow regimes based on Mach number, Ma, and Reynolds number, Re=ρuh/μ. In this work we are interested in flow regimes with low Mach numbers, Ma∼O(ϵ) and low Reynolds number, Re∼O(ϵ), i.e., the flows having Knudsen numbers of O(1), given by the relation:(5)Kn=π2γMaRe,
where γ is the specific heat ratio. In these flow regimes with Kn ∼O(1), the wall-normal velocity and stream-wise gradients of *u* are insignificant as is evident by an order of magnitude analysis [11,25]. Therefore under the assumption of a steady-state fully developed flow, and neglecting the above terms and non-linear terms, the governing equations reduce to the following form: (6)$$\begin{array}{ccc} \frac{\partial (\rho u)}{\partial x} & = & \;0, \end{array}$$(7)$$\begin{array}{ccc} \frac{\partial }{\partial y}\left(\mu \frac{\partial u}{\partial y}\right) & = & \frac{\partial P}{\partial x}, \end{array}$$(8)$$\begin{array}{ccc} \frac{\partial P}{\partial y} & = & \;0. \end{array}$$
The reduced *y*-momentum equation, Equation (8), implies that pressure in the channel is a function of stream-wise coordinate, i.e., P=P(x), and from the reduced continuity and the *x*-momentum equation we can infer the stream-wise velocity, *u*, is a function of coordinates *x* and *y*.

### 2.2. Slip boundary Conditions

In most flow regimes where Kn is quite small (Kn ≤ 0.001), no slip boundary condition is used to describe velocity at the solid interface. The condition assumes that the velocity of the fluid layer in direct contact with the boundary is identical to the velocity of the boundary, i.e.,
(9)$$\begin{array}{c} u_{slip}=u-u_{w}=0, \end{array}$$
where $u_{w}=0$ is the velocity of the stationary solid wall. However, it is now well established that this condition fails to predict the near-wall velocity for rarefied gas flows. In 1879, Maxwell [22] introduced a first order slip formulation widely known as Maxwell slip, which is of the form:(10)$$\begin{array}{c} u_{slip}=K_{s}\lambda \frac{\partial U_{x}}{\partial y}+\frac{3}{4}\frac{\mu }{\rho T}\frac{\partial T}{\partial x}, \end{array}$$
where λ is the mean free path and $K_{s}$ is the standard slip coefficient. The temperature gradient term in Equation (Equation 10) can be neglected under the current assumption of an isothermal flow but the term is important to predict flows arising from thermal gradients at the interface [31]. Researchers have also suggested a slip-velocity dependent on pressure gradients of the form [25,32]:(11)$$\begin{array}{c} \rho \;u_{slip}\propto -\frac{\mu }{P}\frac{\partial P}{\partial x}. \end{array}$$ The definition of slip velocity in Equation (Equation 11) is reminiscence of the diffusion component in the expression of volume-velocity in Recast Navier–Stokes system of equations [20,21]. Conventionally, derivation of continuum flow models as well as their analysis is routinely based on the flow’s mass-velocity (i.e., a velocity definition based on mass-averaging). Recast Navier–Stokes equations (RNS) developed in [20] define three type of volume-velocity variables, $U_{v},\;U_{T}$ and $U_{p}$, that are each a sum of the standard mass-velocity, $U_{m}$, and a diffusion velocity component which takes into account one of the thermodynamic variable, namely, density, temperature or pressure, as:(12)$$U_{v}=U_{m}\;+\;\kappa_{m}\;\nabla \ln\rho =U_{m}\;+\;\frac{\kappa_{m}}{\rho }\nabla \rho ,$$
where $\kappa_{m}$ is a molecular diffusivity coefficient,
(13)$$U_{T}=U_{m}\;+\kappa_{T}\nabla \ln T=U_{m}\;+\frac{\kappa_{T}}{T}\nabla T,$$
where $\kappa_{T}$ is a molecular thermal diffusivity coefficient, and
(14)$$U_{p}=U_{m}\;+\;\kappa_{p}\;\nabla \ln P\;=\;U_{m}\;+\;\frac{\kappa_{p}}{P}\nabla P,$$
where $\kappa_{p}$ is a molecular pressure diffusivity coefficient. In a standard flow simulation a slip or no-slip boundary conditions may be adopted on the mass velocity, $U_{m}$. In this paper we propose to set these boundary conditions using the new velocity variables described in the RNS system. For example, in the pressure-driven flow in a micro- or nano-channel, we set a slip or no-slip on the pressure-diffusion velocity, $U_{p}$, defined in Equation (Equation 14). For a 2-dimensional rectangular channel as detailed in Figure 1, setting a first-order Maxwellian slip boundary condition on the stream-wise volume-velocity $u_{p}$ translates into the following mass-velocity boundary conditions:(15)$$\begin{array}{c} u_{slip}=-\frac{\kappa_{p}}{P}\frac{\partial P}{\partial x}\mp K_{s}K_{b}\frac{\partial U_{x}}{\partial y}\;\mathrm{at}\;y=\pm \frac{h}{2}. \end{array}$$
Here $K_{b}=\lambda$ in the case of gas flow and $K_{s}$ is the standard slip coefficient. Reddy and Dadzie [33] in their work on the effects of molecular diffusivity on shock-wave structures in monatomic gases assumed the following form for the molecular pressure diffusivity coefficient: $$\kappa_{p}=\alpha_{p}\;\mu /\rho .$$
In the present paper, we consider the above relation for the molecular pressure-diffusivity coefficient where the coefficient $\alpha_{p}$ is the phenomenological coefficient determined from the experimental data.

### 2.3. Effects of Rarefaction

Various length scales exist in the evaluation of gas micro-flows. At the molecular level, the mean free path (MFP) is considered along with other characteristic length scales such as mean molecular diameter and mean molecular spacing. In microfluidics where solid boundaries enclose the fluid, the MFP may be smaller than the characteristic length scale of the system and surface effects must be taken into account. The idea of “effective MFP”, $\lambda_{e}$, as a spatially varying function near the wall, can be traced back to the work of Stops [34]. Non-linear constitutive relations may be developed by incorporating effective-transport coefficients such as diffusivity, viscosity and thermal-conductivity based on $\lambda_{e}$, thereby accounting for rarefaction effects in a continuum description [35,36,37,38].

In the present work we adopt the definition of mean free path (λ) provided by G.A. Bird [16]
(16)$$\begin{array}{c} \lambda =\frac{\mu }{P}\sqrt{\frac{\pi }{2}RT}. \end{array}$$

The associated Knudsen (Kn) number for the micro-channel shown in Figure 1, with a characteristic length *h*, pressure *P* and temperature *T* is:(17)$$\begin{array}{c} \mathrm{Kn}:=\frac{\lambda }{h}=\frac{\mu }{hP}\sqrt{\frac{\pi }{2}RT}. \end{array}$$

Beskok and Karniadakis [35] in their rarefaction theory suggested a Bosanquet-type of expression for the viscosity in the transition regime and conducted numerical computations of flow in cylinders and channels using the Navier–Stokes equations complemented with slip boundary conditions. They suggested a Knudsen number dependent effective viscosity with a parameter *a*. Their formulation of the effective viscosity was of the form:(18)$$\begin{array}{c} \mu_{e}:=\mu \frac{\lambda_{e}}{\lambda }=\frac{\mu_{0}}{1+a\mathrm{Kn}}. \end{array}$$

In the present study we adopt the above effective viscosity model in Equation (Equation 18) and assume *a* to be a constant as seen in the works of Michalis [38] and Lv [37]. Using this in our diffusion-slip model we incorporate the rarefaction effects into the molecular pressure diffusivity as:(19)$$\begin{array}{c} \kappa_{p}:=\alpha_{p}\frac{\mu_{e}}{\rho }=\frac{\alpha_{p}}{1+a\mathrm{Kn}}.\frac{\mu_{0}}{\rho } \end{array}$$

### 2.4. Analytical Solution

We solve the reduced governing equations Equations (Equation 6)–(8), along with the following boundary conditions,
(20)$$\begin{array}{ccc} u & = & \;\;\pm K_{s}K_{b}\frac{\partial u}{\partial y}-\frac{\kappa_{p}}{P}\frac{dP}{dx}\;\mathrm{at}\;\;y=\pm \frac{h}{2}, \end{array}$$
(21)$$\begin{array}{ccc} P & = & \;\;P_{in}\;\mathrm{at}\;\;x=0, \end{array}$$
(22)$$\begin{array}{ccc} P & = & \;\;P_{o}\;\mathrm{at}\;\;x=L. \end{array}$$

The velocity profile can be derived as a function of the pressure gradient by solving the *x*-momentum equation satisfying the boundary condition in Equation (Equation 20) as:(23)$$\begin{array}{ccc} u & = & \frac{1}{2\mu }\frac{dP}{dx}(y^{2}-\frac{h^{2}}{4}-\frac{2\mu \kappa_{p}}{P}-K_{s}K_{b}\;h). \end{array}$$

Integrating this expression for the stream-wise velocity along the height of the channel and multiplying with the width, *w*, we get the expression for volumetric flux:(24)$$\begin{array}{ccc} \dot{Q} & = & \int_{-\frac{h}{2}}^{\frac{h}{2}}w\;u\;dy, \end{array}$$(25)$$\begin{array}{ccc} & = & -\frac{wh^{3}}{12\mu }\frac{dP}{dx}(1+\frac{12\mu \kappa_{p}}{Ph^{2}}+\frac{6K_{s}}{h}K_{b}). \end{array}$$
Mass flow rate through the micro-channel is calculated by integrating the density times volumetric flux across the length of the channel,
(26)$$\begin{array}{ccc} \dot{M} & = & \frac{1}{L}\int_{0}^{L}\rho \dot{Q}dx. \end{array}$$

For the case of an incompressible fluid, the density of the fluid is constant, and we substitute the expression $K_{s}K_{b}$ in the equation [see Equation (Equation 25)] with slip-length, $K_{l}$. This gives us
(27)$$\begin{array}{ccc} \dot{M} & = & \frac{\rho }{L}\int_{0}^{L}\dot{Q}dx,\\ & = & \frac{wh^{3}\rho \Delta P}{12\mu L}(1+\frac{12\mu \kappa_{p}}{h^{2}\Delta P}\ln P+\frac{6K_{l}}{h}). \end{array}$$

Following the above procedure, the expression for mass flow rate of incompressible fluid through a cylindrical tube with radius ‘R’ and length ‘L’ can be derived in cylindrical coordinates as,
(28)$$\begin{array}{ccc} \dot{M} & = & \frac{\pi R^{4}\rho \Delta P}{8\mu L}(1+\frac{8\mu \kappa_{p}}{R^{2}\Delta P}\ln P+\frac{4K_{l}}{R}),\\ & = & {\dot{M}}_{HP}(1+\frac{8\mu \kappa_{p}}{R^{2}\Delta P}\ln P+\frac{4K_{l}}{R}), \end{array}$$
where ${\dot{M}}_{HP}$ is the mass flow rate relation predicted by the classical Haigen-Poiseuille law.
Stamatiou et al. [21] uses the RNS system of equations and no-slip condition at the wall to derive an equation for mass flow rate for liquid flow through micro-channel as:(29)$${\dot{M}}_{RNS}=\frac{\pi R^{4}\rho \Delta P}{8\mu L}(1+\frac{8\mu \kappa_{p}}{R^{2}\Delta P}\ln P).$$
It is evident that including effect of first order slip to the above relation gives us the mass flow rate expression derived with the present model, in Equation (Equation 28).

In the case of compressible fluid flowing through nano- and micro-channels, we consider expression for volumetric flow rate, Equation (Equation 25), where molecular diffusivity, $\kappa_{p}$, is substituted as per Equation (Equation 19). The rarefaction effects are included by replacing viscosity with an effective viscosity as given in Equation (Equation 18) and $K_{b}=\lambda$ as $\lambda_{o}\rho_{o}/\rho$, respectively. We then substitute ρ with ideal gas equation, Equation (Equation 4) to get the expression of volumetric flux for the compressible case as,
(30)$$\begin{array}{c} \dot{Q}=-\frac{wh^{3}}{12\mu }\frac{dP}{dx}\left(1+a\mathrm{Kn}\right)\left[1+\frac{24\alpha_{p}}{\pi }\frac{{\mathrm{Kn}}^{2}}{{\left(1+a\mathrm{Kn}\right)}^{2}}+6K_{s}\mathrm{Kn}\right], \end{array}$$
where the Knudsen number, Kn is defined as per Equation (Equation 17). Under the assumption of isothermal flow we also have the following expression relating the pressure and Knudsen number at the outlet ($P_{o},{\mathrm{Kn}}_{o}$) with pressure and Knudsen number inside the channel:(31)$$\begin{array}{c} {\mathrm{Kn}}_{o}\;P_{o}=\mathrm{Kn}\;P. \end{array}$$

We use the expression of volumetric flux for a compressible gas, Equation (Equation 30) and calculate the mass flow rate in the channel with Equation (Equation 26) as,
(32)$$\begin{array}{c} \dot{M}={\dot{M}}_{\mathrm{NS}}\left[1+\frac{48\alpha_{p}}{\pi }\frac{{\mathrm{Kn}}_{o}^{2}}{P^{2}-1}\ln\left(\frac{P+a{\mathrm{Kn}}_{o}}{1+a{\mathrm{Kn}}_{o}}\right)+\frac{2K_{s}{\mathrm{Kn}}_{o}\left(a+6K_{s}\right)}{P+1}+\frac{12aK_{s}{{\mathrm{Kn}}_{o}}^{2}}{P^{2}-1}\ln P\right], \end{array}$$
where $P=P_{in}/P_{o}$ is the ratio of pressures at the inlet and outlet of the channel and ${\dot{M}}_{\mathrm{NS}}$ is the mass flow rate expression for compressible flow through a pipe with no-slip boundary condition, and its expression is given by:$${\dot{M}}_{\mathrm{NS}}=\frac{wh^{3}}{24\mu L\;RT}P_{o}^{2}\left(P^{2}-1\right).$$

To derive the expression for pressure profiles we utilise the fact that mass flux across any arbitrary cross-section of the channel is constant. i.e., ${\dot{M}}_{A}=\rho \dot{Q}=\mathrm{constant}$. Substituting the expression for $\kappa_{p}$ and $\dot{Q}$ from Equation (Equation 30) and rearranging the terms we get a differential equation in pressure of the form,
(33)$$\begin{array}{c} \frac{dP}{dx}(P+B+\frac{C}{P+D}+\frac{E}{P})=-Z=\frac{F}{A}, \end{array}$$
where $A=wh^{3}/\left(12\mu RT\right)$, $B=\left(a+6K_{s}\right){\mathrm{Kn}}_{o}P_{o}$, $C=24\alpha_{p}{\left({\mathrm{Kn}}_{o}P_{o}\right)}^{2}/\pi$, $D=a{\mathrm{Kn}}_{o}P_{o}$, $E=6aK_{s}{\left({\mathrm{Kn}}_{o}P_{o}\right)}^{2}$ and $F=\rho \dot{Q}$.

Integrating the above differential equation for pressure with respect to x, we get a function in pressure:$$\begin{array}{c} f\left(P\left(x\right)\right):=\frac{P{\left(x\right)}^{2}}{2}+BP\left(x\right)+C\ln\left(P\left(x\right)+D\right)+E\ln P\left(x\right)=-Zx+C_{1}. \end{array}$$

In the above expression $Z=(f\left(P_{in}\right)-f\left(P_{o}\right))/L$ and the constant of integration, $C_{1}=f\left(P_{in}\right)$ are calculated from the value of function f(P) at x=0 and at x=L. The pressure profiles as a function of the stream-wise coordinate, P(x), can be calculated numerically by solving the equation,
$$f\left(P\left(x\right)\right)=f\left(P_{in}\right)-\left(f\left(P_{in}\right)-f\left(P_{o}\right)\right)\frac{x}{L}$$
By rearranging Equation (Equation 33) we get the following expression for pressure gradient along the stream-wise direction,
(34)$$\begin{array}{c} \frac{dP}{dx}=-\frac{1}{L}\frac{f\left(P_{in}\right)-f\left(P_{o}\right)}{P+B+\frac{C}{P+D}+\frac{E}{P}}. \end{array}$$

The stream-wise velocity profile for the compressible fluid flow in the channel shown in Figure 1 can be written as a function of Knudsen number by rewriting Equation (Equation 23) to include effective viscosity and rarefaction effects:(35)$$\begin{array}{c} u=\frac{1}{2}\frac{h^{2}}{\mu_{e}}\frac{dP}{dx}\left[\left(\frac{y}{h}+\frac{1}{2}\right)\left(\frac{y}{h}-\frac{1}{2}\right)-\frac{4\alpha_{p}}{\pi }\frac{{\mathrm{Kn}}^{2}}{{\left(1+a\mathrm{Kn}\right)}^{2}}-K_{s}\mathrm{Kn}\right]. \end{array}$$

In the limit of Kn →0, the above velocity expression converges to the classical parabolic expression, but when Kn ≫O(1), the diffusion and slip terms dominate in magnitude and the dependence of the stream-wise velocity on the wall-normal coordinate is reduced, thereby, attaining a plug-flow profile with corresponding slip at the walls,
(36)$$\begin{array}{c} u\approx \frac{1}{2}\frac{h^{2}}{\mu_{e}}\frac{dP}{dx}[-\frac{4\alpha_{p}}{\pi }\frac{{\mathrm{Kn}}^{2}}{{\left(1+a\mathrm{Kn}\right)}^{2}}-K_{s}\mathrm{Kn}]. \end{array}$$

## 3. Results and Discussion

Using the expression for mass flow rate, Equation (Equation 32), derived in the previous section, we can predict the mass flow rates for helium gas in a long micro-channel for the various Knudsen regimes covered in the experiment of Ewart et al. [13], whose conditions are summarised in Table 1. The value for model parameter $K_{s}=1.1466$, is chosen as per the first order slip coefficient reported by Cercignani [39] and Sreekanth [40]. Phenomenological coefficients, $\alpha_{p}=0.3724$ and a=0.4614, are determined by fitting with the experimental data of Ewart et al. [13].

In Figure 2 we compare the present results with the experimental data and previous studies as dimensionless flow rate, G, versus mean Knudsen number, ${\mathrm{Kn}}_{m}$. The non-dimensional flow rate is defined as,
$$G=\dot{M}[\frac{L\sqrt{2RT}}{wh^{2}P_{o}\left(P-1\right)}],$$
and the mean Knudsen number, ${\mathrm{Kn}}_{m}$, is calculated using Equation (Equation 17) at a mean pressure of $(P_{in}+P_{o})/2$.

We observe in Figure 2 that the N-S solution with no-slip boundary conditions and pure diffusion boundary conditions diverge significantly from the experimental data, even in the slip flow regime ($0.001<{\mathrm{Kn}}_{m}<0.1$). The used of Maxwell’s first order slip boundary conditions predicts the mass flow rate up to ${\mathrm{Kn}}_{m}<0.5$. Dadzie and Brenner [30] used a first order slip along with the bi-velocity continuum model as the governing equations to predict the mass flow rate. Their model is able to capture the Knudsen minimum and predicts the mass flow rate up to ${\mathrm{Kn}}_{m}=5$ beyond which the model shows a diverging upward trend. The analytical formulation by Lv et al. [37] which also invokes a bi-velocity formulation shows significant improvement by addressing the rarefaction effects and captures the Knudsen minimum and mass flow rate trend in the whole Kn range. Our present model follows this, matching the experimental data well along the whole range of Knudsen number up to 50 with a better fit than Lv et al. model in the transition regime.

In Figure 3, a comparison is made of normalized stream-wise pressure distributions against the experimental data of Pong et al. [41] and analytical solution for pressure profile provided by Arkillic et al. [11] at various pressure ratios (P). Here, pressure is normalized with the outlet pressure of the micro-channel and the pressure ratio is defined as the ratio of inlet to outlet pressure. The measurements by Pong et al. [41] were made by embedding measurement ports in a micro-channel in which pressure transducers were mounted. The working gas was nitrogen and the outlet Knudsen number was fixed at 0.044. The current model predicts the non-linear pressure profiles very accurately within the errors of the experiment as seen in Figure 3.

The variation of non-linearity in the pressure distribution predicted by the current model can be better understood in the comparison made in Figure 4. In this figure the pressure non-linearity expressed as, (P−Plin)(Pin−Po), measures the deviation from a linear pressure profile from the incompressible flow case, $P_{lin}$. The comparison of pressure profiles are made for various outlet pressures with fixed pressure ratio of P=4. We have considered outlet pressures 0.25$P_{ch}$, 0.5$P_{ch}$, $P_{ch}$, 2$P_{ch}$ and 4$P_{ch}$ which correspond to outlet Knudsen numbers of 4, 2, 1, 0.5 and 0.25, respectively, where $P_{ch}=\frac{\mu }{h}\sqrt{\frac{\pi RT}{2}}$ is pressure at which outlet Knudsen number is 1. There is an apparent asymmetry observed in the pressure profiles in Figure 4. The location of the peaks or dips depend on the inlet/outlet pressure ratio and the Knudsen number. Current slip model predicts that with increase in Kn, the curvature of the nonlinear pressure distribution changes from a convex profile with respect to origin to a concave profile. The change in curvature of pressure profile is also captured by volume diffusion hydrodynamic model [42], second-order slip model [18] and ENSE with a diffusion boundary condition [32]; however, N-S equations with Maxwell’s first-order slip model fails to predict this phenomena.

We use the analytical solution for the stream-wise velocity in Equation (Equation 35) to plot the velocity profiles for a micro-channel at various states of rarefaction. In Figure 5 we see that the current model predicts a parabolic profile with a small slip for Kn = 0.01. As the Knudsen number increases, slip increases until the parabolic profiles with slip at the wall becomes a full plug flow profile in the free molecular regime at Kn = 10. It is evident from Equation (Equation 35) that in the limit of Kn →0, the analytical solution for velocity converges to the description of a parabola as is the case in the continuum regime. The progression of the flow profile from a parabola with slip to a full plug flow is seen in the stream-wise velocity contours in Figure 6.

## 4. Conclusions

In this article, we developed a diffusion-slip model that predicts mass flow rates, velocity distribution and pressure profiles in a rectangular channel over the entire flow regime. The new slip model is a three-parameter model and is able to predict the Knudsen minimum that occurs in the transitional regime. The model compares well against other analytical solutions in predicting the mass flow rates up to free molecular regime.

The proposed diffusion-slip model derived from adopting the definition of other type of velocity variables alongside the conventional mass velocity as seen in the recast Navier–Stokes models [20]. Setting a first order Maxwell’s slip boundary conditions on the new velocity variable leads to the new diffusion-slip model. This investigation strengthens the usefulness of the concept of volume diffusion and the definitions of the new velocity variables. The article also details on the variation of pressure profiles for micro-channels with rarefaction. The present model is able to capture the non-linearity and the change in curvature of pressure profiles with increasing Knudsen number as reported in the literature. A characteristic pressure is determined below which the flow is dominated by diffusion. The velocity profiles predicted by using the new slip model with N-S equations agree with the trends observed in a micro-channel at various states of rarefaction. In the free molecular regime, the diffusive terms have a larger effect in governing the flow profiles, thereby giving a plug flow velocity profile along the channel. This plug flow profile can be noted in the numerical simulations in the works of Christou and Dadzie [43]. The new slip conditions and analytical solutions provide an alternative physical explanation for rarefied flows in micro-channels and may be adopted to interpret other non-equilibrium flow phenomenons.

## Figures and Tables

**Figure 1 micromachines-13-01425-f001:**
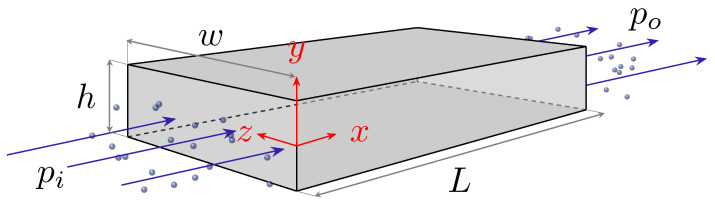
Schematic of pressure driven flow of a rarefied gas through a micro-channel of length *L*, height *h* and width *w* (L,w≫h).

**Figure 2 micromachines-13-01425-f002:**
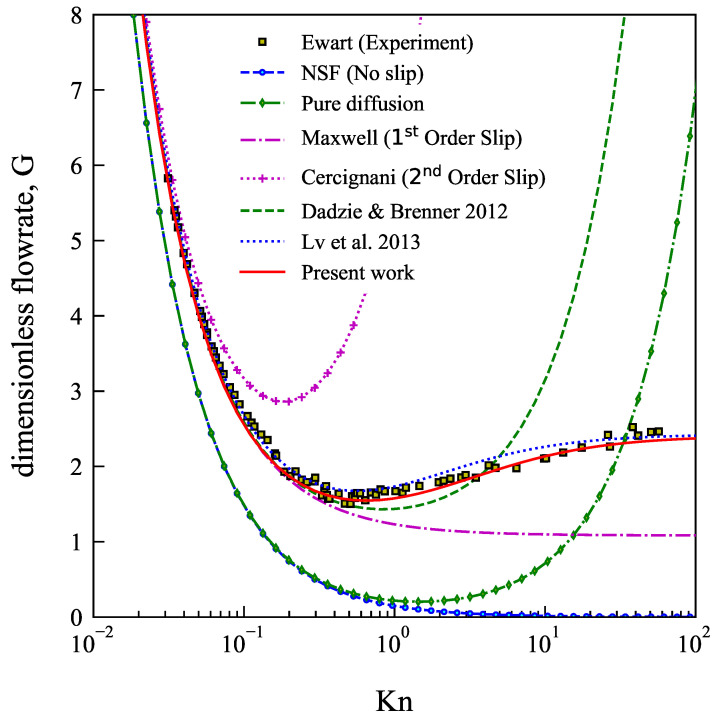
(Color Online) Comparison of dimensionless flow rate, *G*, versus the mean Knudsen number, ${\mathrm{Kn}}_{m}$, between current work and other analytical expressions and the experimental data by Ewart et al. [13,30,37].

**Figure 3 micromachines-13-01425-f003:**
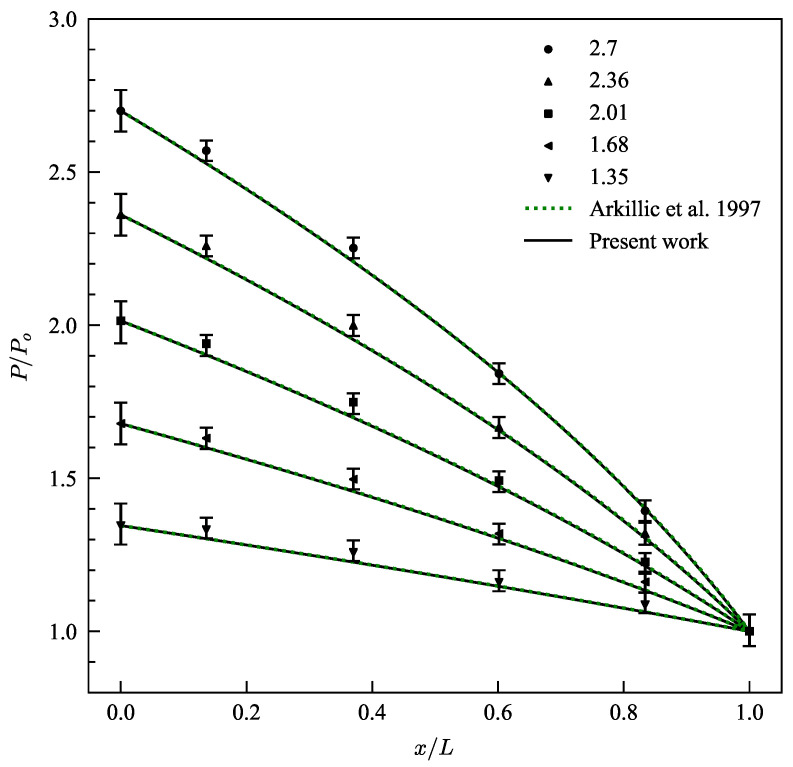
(Color Online) Pressure profiles along the micro-channel for various inlet-outlet pressure ratios as predicted by the current model (solid-line) plotted against the experimental data by Pong et al. [41] and pressure profiles predicted by analytical solution of Arkilic et al. [11].

**Figure 4 micromachines-13-01425-f004:**
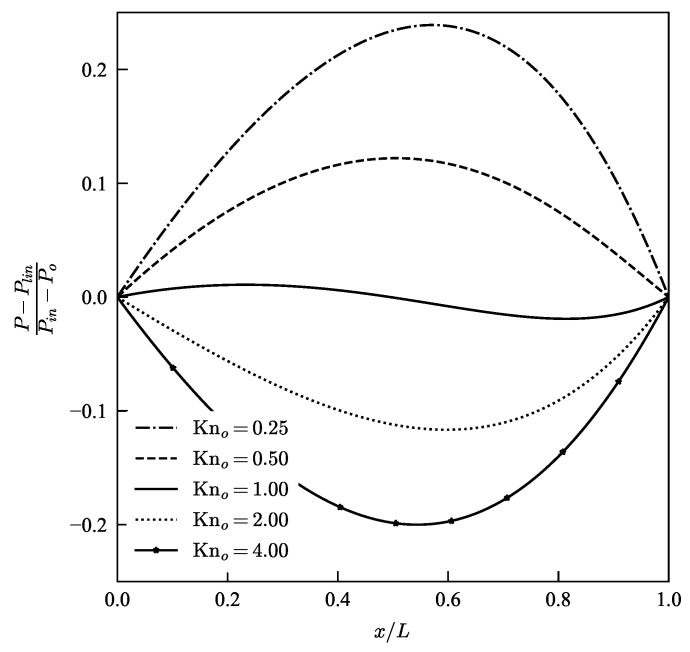
Variation in non-linearity of pressure profiles along the micro-channel for a pressure ratio P=4, plotted for various outlet pressure conditions.

**Figure 5 micromachines-13-01425-f005:**
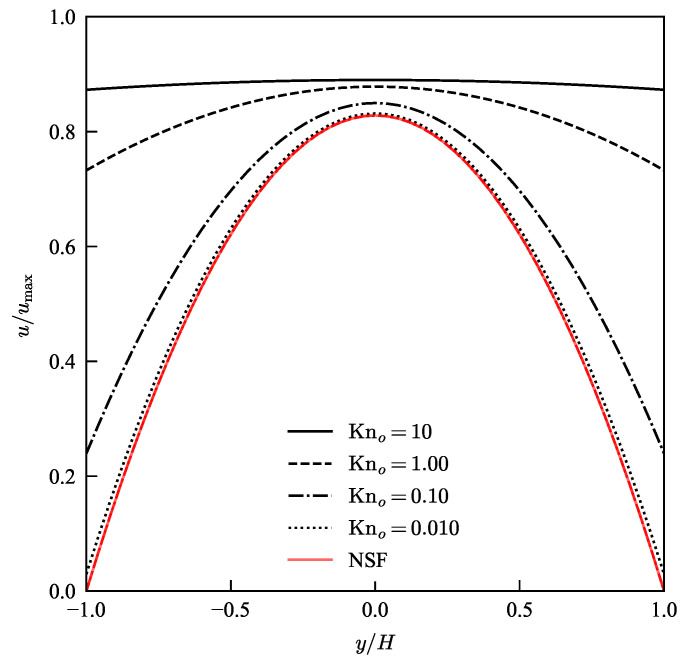
(Color Online) Comparison of normalised stream-wise velocity plotted as a function of normalised height, y/H, at location X = 0.9 along the micro-channel, for various Knudsen numbers as predicted by current model (in black) and N-S with no-slip condition (in red). The velocity profiles are normalised with the corresponding maximum value for each case which corresponds to the exit velocity along the center-line of the pipe.

**Figure 6 micromachines-13-01425-f006:**
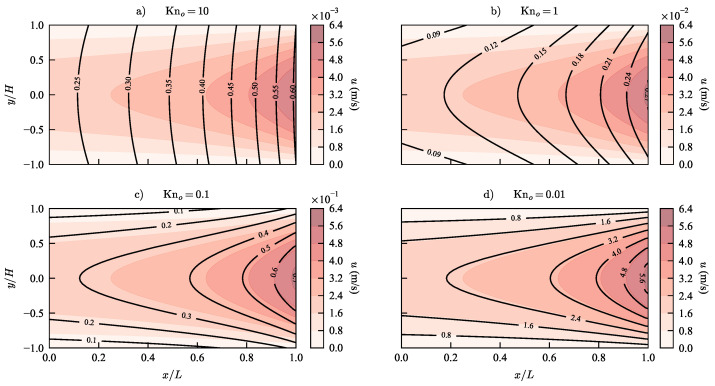
(Color Online) Stream-wise velocity contours predicted by the new model for outlet pressure conditions corresponding to Knudsen numbers (**a**) 10, (**b**) 1, (**c**) 0.1 and (**d**) 0.01. The velocity contour lines are plotted over the stream-wise velocity contour field for a typical parabolic velocity profile predicted by N-S equations with no slip boundary condition.

**Table 1 micromachines-13-01425-t001:** Experimental conditions of Ewart et al. [13] for a pressure ratio of P=5.

Experimental Parameters	Value
Gas used	Helium
Length, *L*	9.39 ± 0.1 mm
Height, *h*	9.38 ± 0.2 μm
Width, *w*	492 ± 1 μm
Avg. Temperature, *T*	296 K
Viscosity, μ	$1.967\times 10^{-5}$ Pa s
Gas Constant, R	2078.5 J/(kg K)
Inlet Pressure range	60.4–109,825 Pa
Outlet Pressure range	12.2–22,633 Pa
Average Kn range	0.027–50.2

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
