# Peer review of "Diffusion-Slip Boundary Conditions for Isothermal Flows in Micro- and Nano-Channels"

_micromachines, 2022, doi:10.3390/mi13091425_

Round 1

Reviewer 1 Report

The authors presented a very interesting approach for the boundary conditions of flows of varied Knudsen numbers. The presented solution is validated by extensive experimental data and seems to be promising for complex numerical models. I suggest some minor corrections to improve the text:

> Equation 5: There seems to be a typo in this equation, the beta should be replaced by gamma pi.

>Paragraph line 248: the sentencing of this paragraph is a bit confusing. Could you clarify the conditions of the several predicted profiles maybe by shortening the longer sentences.
>Same paragraph: Could you comment on the apparent asymetry of the pressure profiles?

Author Response

We thank the reviewer for the comments, please find the response in attached file.

Reviewer 2 Report

In the paper under review, the authors have proposed derivation of model boundary conditions that may systematically justify various diffusion processes occurring in micro- and nano-flows where the classical continuum model breaks down. The authors have concluded that these diffusion-type boundary conditions may be more appropriate to use in simulating flows in micro- and nano-systems and may also be adapted as in interfacial flow modeling.
The subject is important both from the theoretical and engineering points of view. I think that this work will be useful to specialists in fluid dynamics. However, it seems strange that the authors do not try to compare the proposed slip conditions with the classical slip conditions such as Navier slip (1823). One can analyze the Navier slip problem for non-isothermal flow of a viscous fluid in a plane channel (see, e.g., A. A. Domnich et al. A nonlinear model of the non-isothermal slip flow between two parallel plates, J. Phys. Conf. Ser., 1479 (2020), 012005, https://doi.org/10.1088/1742-6596/1479/1/012005). Moreover, I suggest also that the authors analyze such conditions as the free slip condition, threshold slip, etc. (see, e.g., the mini-review [K. R. Rajagopal, On some unresolved issues in non-linear fluid dynamics, Russian Math. Surveys, 58:2 (2003), 319–330, https://doi.org/10.1070/RM2003v058n02ABEH000612]). It is necessary to understand what advantages the new conditions provide in comparison with the classical ones (if there are such advantages) and to explicitly indicate how these advantages can be used in solving practical problems.
Thus, I wish to ask the authors to take into account my comments in order to provide more valuable work for the community.

Author Response

We thank the reviewer for going through our manuscript, please find attached our response in the file attached.

Round 2

Reviewer 2 Report

In their response, the authors clarified some initially unclear points. The article can be recommended for publication.